# Comparison of efficacy and safety of complementary and alternative therapies for essential hypertension with anxiety or depression disorder

Xueyan Han[1]°, Xinxin Liu[1]°, Fengxing Zhong[2], Yiguo Wang[2], Qiming Zhang[1,2]*

**1** Department of First College of Clinical Medicine, Shandong University of Traditional Chinese Medicine, Jinan, Shandong Province, China, **2** Department of Experimental Research Center, China Academy of Chinese Medical Sciences, Beijing, China

° These authors contributed equally to this work.
* zhang_917@126.com

**Funding:** This study is supported by Key R & D Program of the People's Republic of China (2019YFC1711700).QZ received the award. URL=http://www.cncbd.org.cn/. The funders had and will not have a role in study design, data collection and analysis, decision to publish, or preparation of the manuscript.

## Abstract

### Background

Essential hypertension (EH) with anxiety or depression belongs to the category of psycho-cardiology. Hypertension is closely related to anxiety and depression. The adverse reactions of Western medicine are apparent and the compliance is poor. Supplementary and replacement therapies have accumulated rich experience in clinical practices, which can reduce side effects and improve clinical efficacy. This study intends to use the Bayesian network meta-analysis (NMA) analysis method for the first time to gather randomized controlled trials (RCTs) related to complementary and alternative therapies in the treatment of hypertension with anxiety or depression disorder and rank efficacy and safety, to provide a reference basis for the treatment of hypertension with anxiety or depression disorder.

### Methods

All randomized controlled trials (RCTs) and registered and ongoing trials of Chinese and English databases, related to supplementation and replacement therapies of EH with anxiety or depression disorder, published from initial state to February 2021, will be collected in the form of computer retrieval. Two researchers will independently screen the literature, extracting data, assessing bias risk and assessing heterogeneity. We will use software Win-BUGS 1.4.3 and Stata 16.0 for pairwise meta-analysis and NMA to comprehensively evaluate various interventions. The quality of evidence will be evaluated through the Grading of Recommendations Assessment, Development and Evaluation (GRADE).

### Results

This NMA will comprehensively compare and rank the efficacy and safety of a series of complementary and alternative therapies in treating EH with anxiety or depression disorder.

**Competing interests:** The authors have declared that no competing interests exist.

## Conclusion

Supplementary and replacement therapies have accumulated rich experience of clinical practices in improving EH with anxiety or depression disorder. We expect that this NMA will guide practice and research by providing reliable evidence of evidence-based medicine for the treatment of EH with anxiety or depression disorder.

## Protocol registration number

INPLASY202120068.

## Introduction

Psycho-cardiology, which was put forward by James W. Jefferson for the first time [1], mainly studies the relationship between psychological diseases and cardiovascular diseases. Patients with cardiovascular diseases are often associated with anxiety or depression disorder, while hypertension is more likely to occur depression or anxiety among the common and frequently-occurring cardiovascular diseases.

In recent years, the prevalence of essential hypertension (EH) in China has increased. Hypertension is a primary risk factor for high morbidity and mortality of cardiovascular and cerebrovascular diseases worldwide. The World Health Organization has reported that 54% of stroke and 47% of ischemic heart disease are direct consequences of hypertension [2]. Although the epidemiological association between hypertension and cardiovascular disease morbidity/mortality is well known, and there is sufficient evidence to justify antihypertensive treatment, blood pressure (BP) is still not fully controlled. The reason may be the neglect of the intervention of psychological factors. Studies have shown that anxiety and depression are recognized risk factors for the development of EH, which augment the risk of death of cardiovascular disease, especially hypertension, by 4 times [3, 4].

Hypertension is closely related to anxiety and depression [5]. The incidence of hypertension complicated with anxiety and depression is 15–50% [6]. The incidence of depression in patients with hypertension is higher than that in patients with normal BP [7], and about 11.6% of middle-aged and elderly patients with hypertension are complicated with anxiety disorder [8]. Cohen et al. [9] have shown that anxiety can change the circadian rhythm of BP, leading to nocturnal hypertension and morning hypertension. Kayano et al. [10] after monitoring the ambulatory BP of 120 patients with hypertension, it was found that the night and early morning BP of patients with anxiety was higher than patients without anxiety, and the anxiety group was more likely to have anti-dipper BP. A meta-analysis included 9 related articles involving 22367 subjects with an average follow-up period of 9.6 years [11]. The results showed that depressive patients had a 1.42-fold higher risk of developing hypertension than non-depressive patients after adjusting for other risk factors. Depression can lead to elevated BP and then develop into systolic hypertension [12].

The mechanism of anxiety and depression leading to the increase of BP is roughly in 3 aspects. First, anxiety can affect the vascular endothelial function by activating the sympathetic nerve, causing the vascular endothelium to release vasoactive substances out of balance and affect the regulation of BP [5, 13]. Secondly, anxiety and depression can activate the Hypothalamic-Pituitary-Adrenal (HPA) axis system, promoting the secretion of glucocorticoids, improving the sensitivity of vascular smooth muscle to catecholamines, and promote the occurrence and progression of hypertension [14, 15]. Finally, anxiety and depression can

activate the Renin-Angiotensin-Aldosterone System (RAAS), which leads to an increase in BP [16].

EH with anxiety or depression belongs to the category of psycho-cardiology. In terms of treatment, we should not only treat hypertension but also pay more attention to psychological diseases in order to achieve an excellent antihypertensive effect. Western medicine is often combined with anti-anxiety drugs or antidepressants. Although the anxiety and depression of patients are alleviated to some extent, many adverse reactions such as respiratory inhibition, mental fatigue, addiction, gastrointestinal reactions and so on are bad for the quality of life of patients. Another significant problem in treating anti-anxiety drugs or antidepressants is poor compliance, with about 50% of patients discontinuing prescription drugs in the first month of medication [17]. Studies have shown that tricyclic antidepressants and antidepressants that lower adrenaline and serotonin may lead to grade 1 hypertension in patients [18]. Supplementary and replacement therapies have accumulated rich experience in clinical practices, which can reduce side effects and improve clinical efficacy. Yoga, Tai Chi and Qigong are recognized as the complementary approaches for patients with anxiety or depression to improve sleep quality and reduce blood pressure [19]. Some studies have proved that yoga, slow walking, swimming and cycling can effectively relieve symptoms of anxiety and depression [20]. The modified Da Chaihu decoction was beneficial for hypertensive patients with anxiety in reducing BP [21]. Brief mindfulness meditation could significantly reduce anxiety symptoms and lower systolic blood pressure of Chinese nursing students [22].

The network meta-analysis (NMA) can be used to compare three or more interventions simultaneously, and can provide the probability of choosing the best intervention. This study intends to use the Bayesian network meta-analysis (NMA) analysis method for the first time to gather randomized controlled trials (RCTs) related to complementary and alternative therapies in the treatment of hypertension with anxiety or depression disorder and rank efficacy and safety, to provide a reference basis for the treatment of hypertension with anxiety or depression disorder.

## Methods

This NMA will be implemented according to the Preferred Reporting Items for Systematic Review and Meta-Analysis Protocols (PRISMA-P) details [23]. The protocol was registered on the International Platform of Registered Systematic Review and Meta-analysis Protocols (INPLASY). No: INPLASY202120068.

### Ethics and dissemination

This study is based on published data, and so it does not require ethical approval. The results of this NMA will be presented at scientific conferences and published in a peer-review journal.

### Inclusion criteria

**Type of study.** The type of study is RCTs related to complementary and alternative therapies for EH complicated with anxiety or depression disorder, and the languages are Chinese and English.

**Participants.**

1. Patients who have diagnosed EH (SBP≥140 mmHg or DBP≥90mmHg, according to the diagnostic criteria of the Guidelines for Prevention and Control of Hypertension in 2010), with anxiety or depression disorder (according to the score of any validated scales).

2. It is not affected by the age, sex and region of the included patients.

**Interventions.** The control group was treated with or without conventional hypotensive drugs, at the same time, with or without anti-anxiety drugs or antidepressants. The treatment group was treated with complementary replacement therapies on the basis of the control group. The complementary replacement therapies include biological-based therapies, traditional Chinese medicine (TCM), acupuncture, massage, qigong, moxibustion, yoga, Chinese herbal medicines, music therapy, five-animal exercises, cognitive-behavioral therapy, relaxation training, tai chi, mindfulness meditation, and so on. They can also be used without conventional hypotensive drugs.

**Outcomes.** Primary outcomes include the change of SBP and DBP, the change in score of any validated scales, such as HAMA, HAMD, or SAS, SDS, which can assess the severity of anxiety or depression. The total efficacy rate, stationarity of hypotension, adverse effects, and TCM symptoms score are secondary outcomes.

## Exclusion criteria

1. The type of study is non-RCTs;

2. The subjects included were diagnosed patients with secondary hypertension;

3. Patients without anxiety or depression;

4. Studies of unknown specific inclusion, exclusion criteria, and intervention measures;

5. There are obvious errors or the research data are unknown and cannot be extracted.

## Databases and retrieval strategy

Chinese and English databases such as MEDLINE, PubMed, Embase, ClinicalTrials.gov, PsycINFO, Web of Science, Cochrane Library, Biosis, CNKI, Wanfang Database, Weipu Database and China Biology Medicine disc (CBMdisc) and so on will be searched by two independent researchers. RCTs, related to supplementation and replacement therapies of EH with anxiety or depression disorder, which were published from initial state to February 2021, will be collected in the form of computer retrieval. For the researches on incomplete information or some problems in the data, the corresponding author will be contacted to supplement the missing data as much as possible. At the same time, references of systematic review/meta-analysis as well as registered and ongoing trials will be traced. Take PubMed as an example, and the retrieval strategy is shown in Table 1.

## Literature screening and data extraction

Literature screening: according to the retrieval strategy, the two researchers will check all the databases independently, and obtain all the studies related to this research. Then, by reading the title and abstract of studies, the studies which have nothing to do with the research will be excluded. Finally, we will read through the full text and screen the literature according to the inclusion and exclusion criteria mentioned above. The data will be imported into Endnote X9 software (Clarivate Analytics, Philadelphia, PA USA).

Data extraction: the chart of literature information will be drawn, and the two researchers will independently extract the information from the included researches. The level of agreement between the two researchers will be analysed by using the Kappa value [24], and if the value is greater than 0.60, it means substantial/almost perfect agreement. The extracted information will include: title, first author, race, gender, sources of participants, course of disease, publication time, country, journal, the support of the fund, the sources of the original literature, the number of cases in each group, inclusion and exclusion criteria, intervention

**Table 1. Search strategy for PubMed.**

| NO. | Search item |
|---|---|
| #1 | Essential Hypertension [MeSH] |
| #2 | Essential Hypertension[Title/Abstract] OR High Blood Pressure[Title/Abstract] OR Hypertension[Title/Abstract] OR Blood Pressure[Title/Abstract] OR Psycho-cardiology[Title/Abstract] OR Psychocardiology[Title/Abstract] |
| #3 | #1 OR #2 |
| #4 | Depression[MeSH] |
| #5 | Depressions[Title/Abstract] OR Emotional Depressions[Title/Abstract] OR Depression, Emotional [Title/Abstract] OR Depressive Symptom[Title/Abstract] OR Symptom, Depressive[Title/Abstract] OR Depressions, Emotional [Title/Abstract] OR Emotional Depression[Title/Abstract] OR Depressive Symptoms [Title/Abstract] OR Symptoms, Depressive [Title/Abstract] OR Depression Disorder [Title/Abstract] |
| #6 | #4 OR #5 |
| #7 | Anxiety[MeSH] |
| #8 | Anxiety[Title/Abstract] OR Anxiety Disorder[Title/Abstract] OR Emotional Anxieties[Title/Abstract] OR Anxiety, Emotional[Title/Abstract] OR Symptom, Anxious[Title/Abstract] OR Anxieties, Emotional[Title/Abstract] OR Emotional Anxiety[Title/Abstract] OR Anxious Symptom[Title/Abstract] OR Symptoms, Anxious [Title/Abstract] OR Anxious Symptoms [Title/Abstract] |
| #9 | #7 OR #8 |
| #10 | #3 AND #6 |
| #11 | #3 AND #9 |
| #12 | #10 OR #11 |
| #13 | Complementary Therapies[MeSH] |
| #14 | Therapy, Alternative[Title/Abstract] OR Complementary and Alternative Therapies[Title/Abstract] OR Complementary Medicine[Title/Abstract] OR Alternative Therapies[Title/Abstract] OR Medicine, Complementary[Title/Abstract] OR Alternative Medicine[Title/Abstract] OR Complementary Therapies [Title/Abstract] OR Medicine, Alternative[Title/Abstract] OR Therapy, Complementary[Title/Abstract] |
| #15 | #13 OR #14 |
| #16 | Chinese Herbal Medicines[Title/Abstract] OR Yoga[Title/Abstract] OR Exercise[Title/Abstract] OR Music Therapy[Title/Abstract] OR Chinese Medicine[Title/Abstract]OR Biological-based Therapies[Title/Abstract] OR Five-Animal Frolics Exercise[Title/Abstract] OR Acupuncture[Title/Abstract] OR Massage[Title/Abstract] OR Cognitive-behavioral Therapy[Title/Abstract] OR Relaxation Training[Title/Abstract] OR Traditional Chinese Medicine[Title/Abstract] OR Tai chi[Title/Abstract] OR Qigong[Title/Abstract] OR Moxibustion[Title/Abstract] OR Massage[Title/Abstract] OR Mindfulness Meditation[Title/Abstract] |
| #17 | #15 OR #16 |
| #18 | Randomized Controlled Trial[Publication Type] OR Controlled Clinical Trial[Publication Type] OR Randomized[Title/Abstract] OR Randomly[Title/Abstract] |
| #19 | #12 AND #17 AND #18 |

measures and implementation details, primary and secondary outcomes, follow-up period, method of randomized allocation, allocation concealment, blinding, loss of follow-up or withdrawal, security, ethics, etc.

## Assessment of trial quality

The quality of literature will be scored according to the Jadad scale [25]. The Jadad score of the selected literature should be greater than or equal to 3. In case of disagreement, consult the third party to decide.

## Statistical analysis

### Pairwise meta-analysis

Stata 16.0 software will be used for statistical analysis. The ratio (OR) will be used as the index for the counting data, and the mean difference (MD) will be used for the measurement data.

The 95% confidence interval (95% CI) of each effect size will be given. A $p$-value <0.05 will be considered significant. We will combine with $I^2$ to quantitatively judge the size of heterogeneity.

## Network meta-analysis

In this paper, the NMA is mainly based on the Bayesian theory to compare all the intervention measures and make a comprehensive evaluation at the same time. We will use $I^2$ or $p$–value to determine the types of model. If $I^2 > 50\%$ or $p \leq 0.05$, it indicates that there is heterogeneity, and we will use the random-effect model. If $I^2 \leq 50\%$ or $p > 0.05$, we will use the fixed-effect model. When there is heterogeneity, we will probe the sources of heterogeneity through sensitivity analysis or subgroup analysis. Markov chain Monte Carlo (MCMC) in WinBUGS1.4.3 is a method of the joint posterior distribution by constructing simulation parameters of Markov-chain. The steps are as follows: checking model logic and syntax, importing data, and compiling. In this study, three chains will be selected for simulation, with 40000 iterations and 10,000 annealing times. Gelman-Rubin-Brooks will be used to evaluate the convergence of the model. The higher the surface under the cumulative ranking curve values (SUCRA) indicates that the better the efficacy of the intervention [26]. And then the efficacy of the intervention measures included in the end will be ranked.

## Subgroup analysis and sensitivity analysis

We will divided data into smaller units according to different design scheme, quality of literature, publishing year, etc., to make a further comparison.

After excluding the studies with abnormal results, the NMA will be carried out again, and the results are compared with the results of NMA, which does not exclude the studies with abnormal results, so as to probe the influence of the studies on the merger effect. The methods of sensitivity analysis include including or excluding those controversial studies on whether they meet the inclusion criteria, excluding studies with low quality, using different statistical methods to re-analyze data, and so on.

## Assessment of inconsistency

IF and 95% CI will be calculated. The consistency of closed loops will be evaluated by 95% CI. We can also use the node splitting method to analyze each node of NMA and compare the difference of direct and indirect comparison. If P > 0.05, it means that there is no inconsistency.

## Evaluation of publication bias and evidence quality

The included data will be tested by Begg´s and Egger´s funnel plot to evaluate whether there are publication bias and the effect of small sample size. If the included studies are concentrated near the midline and symmetrically distributed, it shows that the publication bias has little influence on this research. On the contrary, there will be publication bias.

The quality of evidence is divided into five levels—risk of bias, indirectness, inconsistency, imprecision, and publication bias—in NMA according to Grading of Recommendations Assessment, Development and Evaluation (GRADE) [27]. GRADE is used to evaluate the level of evidence and recommendation intensity.

## Discussion

There is often a causal relationship between hypertension and psychological disorders. The prevalence rate of hypertension in patients with anxiety or depression is significantly higher

than that in patients without anxiety or depression. On the contrary, the risk of anxiety and depression in patients with hypertension is significantly higher than that in patients without hypertension [28–30]. It plays a vital role in each other's occurrence and development, which increases the difficulty in the treatment of hypertension [31, 32].

At present, It is difficult to achieve a satisfactory antihypertensive response by conventional hypotensive drugs for EH with anxiety or depression disorder and the side effects are obvious. Complementary and replacement therapies of EH with anxiety or depression disorder play an increasingly vital role. However, there are many kinds of complementary and replacement therapies and the efficacy is uneven, so it is difficult to choose the best scheme in the clinic. This will be the first NMA to systematically evaluate the efficacy and safety of different complementary and alternative therapies for EH with anxiety or depression disorder, and then we will rank them in order to get the best regimen, which can guide the clinical treatment.

We expect that this NMA will guide practice and research by providing reliable evidences of evidence-based medicine for treatment of EH with anxiety or depression disorder. Nevertheless, this study also has some limitations. For example, the use of different anxiety and depression scales in different studies may lead to publication bias.

## Supporting information

**S1 Checklist. PRISMA-P checklist.**
(DOC)

## Author Contributions

**Conceptualization:** Xueyan Han, Xinxin Liu, Qiming Zhang.

**Formal analysis:** Xueyan Han, Xinxin Liu.

**Methodology:** Fengxing Zhong, Yiguo Wang.

**Project administration:** Qiming Zhang.

**Software:** Xueyan Han, Fengxing Zhong.

**Validation:** Xinxin Liu, Yiguo Wang.

**Writing – original draft:** Xueyan Han, Xinxin Liu.

**Writing – review & editing:** Xueyan Han, Qiming Zhang.

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
