## [Decision Letter · Decision Letter 0]

20 May 2021

PONE-D-21-05843

Comparison of efficacy and safety of complementary and alternative therapies for essential hypertension with anxiety or depression disorder

A Bayesian network meta-analysis protocol

PLOS ONE

Dear Dr. Zhang,

Thank you for submitting your manuscript to PLOS ONE. After careful consideration, we feel that it has merit but does not fully meet PLOS ONE’s publication criteria as it currently stands. Therefore, we invite you to submit a revised version of the manuscript that addresses the points raised during the review process.

We look forward to receiving your revised manuscript.

Kind regards,

Ismaeel Yunusa, PharmD, PhD

Academic Editor

PLOS ONE

Additional Editor Comments (if provided):

Please submit a revised version addressing reviewer comments based on guidance from PRISMA-P (Preferred Reporting Items for Systematic review and Meta-Analysis Protocols). Resubmit along with PRISMA-P checklist

Journal Requirements:

2) We note you have included a table to which you do not refer in the text of your manuscript. Please ensure that you refer to Table 1 in your text; if accepted, production will need this reference to link the reader to the Table.

Reviewers' comments:

Reviewer's Responses to Questions

**Comments to the Author**

1. Does the manuscript provide a valid rationale for the proposed study, with clearly identified and justified research questions?

Reviewer #1: Partly

2. Is the protocol technically sound and planned in a manner that will lead to a meaningful outcome and allow testing the stated hypotheses?

Reviewer #1: Yes

3. Is the methodology feasible and described in sufficient detail to allow the work to be replicable?

Reviewer #1: No

4. Have the authors described where all data underlying the findings will be made available when the study is complete?

Reviewer #1: Yes

5. Is the manuscript presented in an intelligible fashion and written in standard English?

Reviewer #1: Yes

6. Review Comments to the Author

You may also provide optional suggestions and comments to authors that they might find helpful in planning their study.

Reviewer #1: Dr. Xueyan Han and co-workers aimed to perform a systematic review and Bayesian network meta-analysis (NMA) method for the first time to gather randomized controlled trials (RCTs) related to complementary and alternative therapies in the treatment of hypertension with anxiety or depression disorder and rank efficacy and safety, to provide a reference basis for the treatment hypertension with anxiety or depression disorder. I have several concerns.

1. In the results of the Abstract, the authors wrote that “This NMA will comprehensively compare and rank the efficacy and safety of a series of complementary and alternative therapies in treating EH with anxiety or depression disorder”. Could the authors provide how many methods for therapy in treating EH with anxiety or depression disorder were planned to compare?

2. In Literature screening and data extraction of methods, could the authors consider providing the Kapa value between the two researchers?

3. In subgroup analysis and sensitivity analysis of Methods, the authors wrote that "the methods of sensitivity analysis include excluding unpublished studies………." but this could be erroneous to do, because the authors included studies that were published from the initial state to February 2021. Please revise.

4. In network meta-analysis of statistical analysis, the authors didn’t write what model was used to synthesis the data while the MCMC method was just a method used to estimate the unknown parameters. Please revise.

5. In the section on publication bias, the authors didn’t write which method was planned to use for testing the publishing bias. Please revise.

7. PLOS authors have the option to publish the peer review history of their article (what does this mean?). If published, this will include your full peer review and any attached files.

Reviewer #1: No

---

## [Author Response · Author response to Decision Letter 0]

30 Jun 2021

Thank you so much for your time and efforts! We profoundly appreciate your comments toward the improvement of the paper and we hope what we have addressed the majority of your comments. Responses to your comments along with a description of the changes made on the manuscript are given below.

- Additional Editor Comments (if provided)

Point. Please submit a revised version addressing reviewer comments based on guidance from PRISMA-P (Preferred Reporting Items for Systematic review and Meta-Analysis Protocols). Resubmit along with PRISMA-P checklist

Response: Thank you very much for your carefulness and comment! We revised PRISMA-P addressing reviewer comments based on guidance from PRISMA-P and resubmitted a revised version of PRISMA-P.

- Journal Requirements:

Point 1. Please ensure that your manuscript meets PLOS ONE's style requirements, including those for file naming. The PLOS ONE style templates can be found at

Response 1: Thank you very much for your suggestion! We carefully checked PLOS ONE's style requirements, including those for file naming, to make sure that our manuscripts met the requirements.

Point 2. We note you have included a table to which you do not refer in the text of your manuscript. Please ensure that you refer to Table 1 in your text; if accepted, production will need this reference to link the reader to the Table.

Response 2: Thank you very much for your carefulness and comment! We have made changes to ensure that Table 1 is explicitly referenced in the manuscript. "Take PubMed as an example, and the retrieval strategy is shown in Table 1." In line 169-170.

- Reviewer #1

Point 1. In the results of the Abstract, the authors wrote that “This NMA will comprehensively compare and rank the efficacy and safety of a series of complementary and alternative therapies in treating EH with anxiety or depression disorder”. Could the authors provide how many methods for therapy in treating EH with anxiety or depression disorder were planned to compare?

Response 1: Thank you very much for your comments on our manuscript! For essential hypertension with anxiety or depression disorder, on the basis of conventional antihypertensive drugs, complementary and alternative therapies, such as Chinese herbal medicines, yoga, music therapy, biological-based therapies, five-animal frolics exercise, acupuncture, massage, cognitive-behavioral therapy, relaxation training, traditional Chinese medicine, tai chi, qigong, moxibustion, mindfulness meditation, etc., can be compared with antihypertensive drugs combined (or not combined) with western medicines with anti-anxiety or anti-depression. They can also be used without conventional hypotensive drugs. These complementary and alternative methods can be found in interventions of inclusion criteria of Methods.

Point 2. In literature screening and data extraction of methods, could the authors consider providing the Kapa value between the two researchers?

Response 2: Thank you very much for your suggestion! In literature screening and data extraction of methods, we think that providing the Kapa value between the two researchers is more objective and reasonable, and we have decided to adopt your suggestion. For the revised details, please refer to the line 182-184 of the revised manuscript. "The level of agreement between the two researchers will be analysed by using the Kappa value, and if the value is greater than 0.60, it means substantial/almost perfect agreement."

Point 3. In subgroup analysis and sensitivity analysis of Methods, the authors wrote that "the methods of sensitivity analysis include excluding unpublished studies………." but this could be erroneous to do, because the authors included studies that were published from the initial state to February 2021. Please revise.

Response 3: Thank you very much for your carefulness and comment! We are sorry for our carelessness. We have deleted "excluding unpublished studies".

Point 4. In network meta-analysis of statistical analysis, the authors didn’t write what model was used to synthesis the data while the MCMC method was just a method used to estimate the unknown parameters. Please revise. 

Response 4: Thank you very much for your carefulness and comment! Considering your comment, we have further improved this part. "We will use I2 or p–value to determine the types of model. If I2 > 50% or p ≤ 0.05, it indicates that there is heterogeneity, and we will use the random-effect model. If I2 ≤ 50% or p > 0.05, we will use the fixed-effect model. When there is heterogeneity, we will probe the sources of heterogeneity through sensitivity analysis or subgroup analysis." In line 204-208.

Point 5. In the section on publication bias, the authors didn’t write which method was planned to use for testing the publishing bias. Please revise.

Response 5: Thank you very much for your carefulness and comment! Considering your comment, we have improved the content of evaluation publication bias. "The included data will be tested by Begg´s and Egger´s funnel plot to evaluate whether there are publication bias and the effect of small sample size. If the included studies are concentrated near the midline and symmetrically distributed, it shows that the publication bias has little influence on this research. On the contrary, there will be publication bias." For the revised details, please refer to the line 232-236 in the revised manuscript.

---

## [Editor Report · Decision Letter 1]

2 Jul 2021

Comparison of efficacy and safety of complementary and alternative therapies for essential hypertension with anxiety or depression disorder

A Bayesian network meta-analysis protocol

PONE-D-21-05843R1

Dear Dr. Zhang,

We’re pleased to inform you that your manuscript has been judged scientifically suitable for publication and will be formally accepted for publication once it meets all outstanding technical requirements.

Kind regards,

Ismaeel Yunusa, PharmD, PhD

Academic Editor

PLOS ONE
---

## [Editor Report · Acceptance letter]

7 Jul 2021

PONE-D-21-05843R1 

Comparison of efficacy and safety of complementary and alternative therapies for essential hypertension with anxiety or depression disorder
A Bayesian network meta-analysis protocol 

Dear Dr. Zhang:

I'm pleased to inform you that your manuscript has been deemed suitable for publication in PLOS ONE. Congratulations! Your manuscript is now with our production department. 

Kind regards, 

on behalf of

Dr. Ismaeel Yunusa 

Academic Editor

PLOS ONE